# Effects of short-term pre-competition weight loss on certain physiological parameters and strength change in elite boxers

Yavuz Yasul[1], Faruk Akçınar[2], Muhammet Enes Yasul[3], Ahmet Kurtoğlu[4], Özgür Eken[5]*, Georgian Badicu[6]*, Luca Paolo Ardigò[7]

1 Bafra Vocational School, Ondokuz Mayıs University, Samsun, Turkiye, 2 Department of Coaching Education, Faculty of Sport Sciences, Inonu Mayıs University, Malatya, Turkiye, 3 Başakşehir Çam ve Sakura City Hospital, Nutrition and Diet Unit, Istanbul, Turkiye, 4 Department of Coaching Education, Faculty of Sport Science, Bandirma Onyedi Eylul University, Balikesir, Türkiye, 5 Department of Physical Education and Sport Teaching, Faculty of Sport Sciences, Inonu University, Malatya, Türkiye, 6 Department of Physical Education and Special Motricity, Faculty of Physical Education and Mountain Sports, Transilvania University of Braşov, Braşov, Romania, 7 Department of Teacher Education, NLA University College, Oslo, Norway

* georgian.badicu@unitbv.ro (GB); ozgureken86@gmail.com (OE)

## Abstract

### Background

Athletes in certain sports aim to gain an advantage by competing in a lower body mass class instead of competing in their own body mass class. This study aims to reveal certain physiologic and strength changes in elite male boxers who lost body mass rapidly before the competition.

### Methods

30 thirty boxers who were aged between 19–24 years and having a mean age of 7.4 years participated in the study. To evaluate the effect of short-term dietary intake interventions on body composition and muscle strength before the competition, boxers were divided into three groups: control (C), exercise+diet1 (E+D1) and exercise+diet2 (E+D2) groups. The dietary habits of the participants were controlled and they participated in the training program. The data of the study consisted of variables such as body mass, height, regional muscle mass, body fat percentage, biceps and femur bicondylar circumference measurements before the competitions. Isometric strength measurements of knee extensors and flexors and shoulder internal and external rotators were also recorded.

### Results

Physiologic parameters such as body mass change, BMI level, body fat percentage and leg muscle ratios of E+D2 were significantly decreased compared to C and E+D1 groups. Furthermore, submaximal and maximal strength production in knee extensors and flexors as well as shoulder internal and external rotators were significantly decreased in E+D2 compared to C and E+D1 groups.

**Data Availability Statement:** All relevant data are within the manuscript and its Supporting Information files, please see "Other".

**Funding:** The author(s) received no specific funding for this work.

**Competing interests:** The authors have declared that no competing interests exist.

## Conclusion

The tendency to lose body mass quickly in a short of time may give the desired results in terms of BMI, body mass and fat percentage, but it may cause strength losses in boxers during the competition period.

## Introduction

Sporting success in boxing depends on the perfect harmony of movements and the coordination that will provide this harmony, that is, a level training routine. Boxers can increase their endurance, speed and strength levels with the right training routines [1]. However, in addition to the right training routines, the development of these abilities is also related to the efficiency and availability of energy stores.

Boxers use mostly aerobic energy metabolism during the fight while practicing this sport. This energy metabolism contributes to the boxers to perform rapid, powerful and repetitive attacks throughout the entire fight [2]. Furthermore, the aerobic energy production pathway promotes the replenishment of high-energy phosphate stores in addition to efficient recovery between rounds and during short resting periods [3]. Therefore, the anaerobic metabolic pathway may provide stronger responses to supply the energy for short and intense attacks with maximal power [4].

Within this framework, energy stores need to be adequately replenished to elicit strong responses and to maintain uninterrupted energy production. The Academy of Nutrition and Dietetics, Dietitians of Canada (DC) and the American College of Sports Medicine (ACSM) emphasized that all processes, from recovery to performance improvement especially in sporting activities, can be improved with well-chosen nutritional strategies [5]. In one aspect, performance is the force generated by physiological tension during muscle contraction. Therefore, a correct nutritional strategy with a correct strength development process is invaluable in terms of revealing the expected power output both in preparation for the competition and during the competition periods. However, many athletes in combat sports wish to gain a body mass advantage by losing body mass rapidly in the pre-competition period [6].

The lack of knowledge about ideal body mass loss in combat athletes and the desire for quick results often have a negatively impact on athletic performance and can also impair athletes' physical health [7]. Due to body mass loss, athletes may experience decreases in power output and endurance levels [8]. In addition, there may be a decrease in metacognitive abilities and weaknesses in attention and decision-making [9]. Moreover, in combat sports, the reaction times and hit accuracy rates of competitors who chose to lose more than 5% of their body body mass were negatively affected [10]. Therefore, it seems to be a better choice at this stage to lose body mass gradually over a few weeks in competitions or qualifiers that last a few days [11]. In the case of boxers, this has not been investigated in depth.

Based on all this information, the hypothesis of this research was based on the change in body composition of boxers whose diets were controlled and calorie-restricted in the pre-competition period and how this change would affect both knee extensors and flexors and shoulder internal and external rotator muscle strength.

## Methods

### Study design

This research is an experimental study with a control group. The study was conducted with 30 boxers born in 2000–2004 who were competitors (the minimum requirements included

**Table 1. 30-day training practice of the boxers before the championship.**

| Warm-up min/hr | | Technical-Tactical min/hr | | Agility | | Cool-down | |
|---|---|---|---|---|---|---|---|
| 20 min (dynamic warm-up series) | shadow boxing and jumping rope, pad work and paired bob and weave exercise, shadow boxing and jumping rope with the coach, and foot stepping. | 40 min (3x10 min, 1 min rest) | Technical shadow boxing with the coach, conditional boxing, free boxing, pad work and shadow boxing | 15 min (3x4, 1 min rest) | Stationary exercise, conditional punching bag, plyometric training, medicine ball training, interval exercise | 15 min | aerobic running, stretching exercises |

participating in the Turkish championship or participating in national competitions). The sample size was determined by G-Power. According to this, taking type I error (alpha) 0.05, power (1-beta) 0.8 and effect size 0.63, the minimum sample size required to detect a significant difference using this test was determined as at least 10 in each group (30 in total). All athletes were taken to a camp to follow the housing and nutrition processes in a controlled manner. The athletes were informed about the measurements to be taken and informed consent form commitments were obtained. The participants were divided into 3 groups (10 participants in each group), control (C), exercise+diet1 (E+D1) and exercise+diet2 (E+D2) groups. The control group was fed as E+D1 and did not participate in training. The E+D1 and E+D2 groups were controlled in terms of both training and dietary habits. The E+D1 and E+D2 group performed the training practices in Table 1 and followed the nutrition program in Table 2. The E+D1 group was planned to participate in the championship as half middleweight and E+D2 as half welterweight. The mean weight of the boxers participating in the study was 70.4. However, within the scope of the championship, the boxers who would compete in the half middleweight division had to weigh 69 kg and the boxers who would compete in the half welterweight division had to weigh 63.5 kg. Therefore, for the E+D2 group to compete in the half-welterweight division, a daily dietary restriction of 500 kcal was adopted. A brief overview of diet and exercise has been presented in Fig 1. Also the research was conducted according to the Declaration of Helsinki guidelines and was approved by the Ondokuz Mayıs University (OMU) and the Human Research Ethics Committee of OMU (approval number: 2023 /90). Informed consent was obtained from all subjects (one for each group; written) agreed to participate in this study and answered the questionnaire.

## Measurement of anthropometric method

On the first day of the camp, body mass measurements were taken in the morning on an empty stomach barefoot and wearing only shorts at 8:00 a.m. via a scale accurate up to ±20g

**Table 2. 30-day diet plan before the championship.**

| | E+D1 Sample | E+D2 Sample |
|---|---|---|
| **Morning** | 1 glass of milk (200 ml), Oats Omelet (4 tablespoons of oat+2 eggs+1 small spoon of olive oil), 1 slice of white cheese (30 gr), 1 apple, 3 thin slices of bread (75 gr) | Tea (sugar-free), 1 boiled egg, 2 slices of white cheese (60 gr), 10 olives, Vegetable salad, 1 medium apple, 3 thin slices of bread (50gr) |
| **Noon** | 1 ladle of soup, Grilled chicken (90 gr), 4 tablespoons of boiled vegetables, 4 tablespoons of yogurt, 3 thin slices of bread (75 gr), 10 raw almonds | 1 ladle of soup, 4 tablespoons of boiled vegetables, 3 tablespoons of yogurt, 1 orange, 3 thin slices of bread (75 gr) |
| **Evening** | 2 ladles of soup, Grilled chicken (90 gr), 3 tablespoons of pilaf, 4 tablespoons of boiled vegetables, 300 ml ayran, 3 thin slices of bread (75 gr) | 1 ladle of soup, Grilled chicken (90 gr), 6 tablespoons of pilaf, 4 tablespoons of vegetables with olive oil, 300 ml ayran, 3 thin slices of bread (75 gr) |
| **Before Exercise** | 1 large banana + 6 dates + half a teacup of roasted chickpeas | 1 large banana + 3 dates |

## Training and diet intervention

The pre-championship training sessions were designed for an average of 90 minutes and consisted of running, technical-tactical, quickness and cooling exercises in Table 1.

Food consumption records were taken for 3 days to determine the dietary routines of the participants. Energy requirement was calculated as basal metabolic rate (BMR)+thermic effect of food (TEF) +physical activity and the daily nutritional requirement was calculated as 2778.9 kcal [12, 13].

The control group did not have any nutrition and training programme. C group was not intervened in normal daily eating habits.

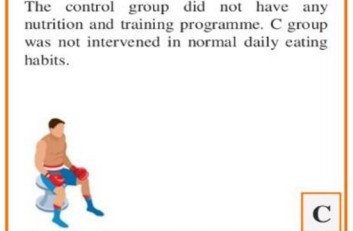

**C**

This group was not subjected to calorie restriction and their diet was adjusted by calculating the amount of protein in the diet as 1.5 g per kilogram. Therefore, 105 g protein (for a 70 kg individual) was added to the 2778.9 kcal diet [12] in Table 2.

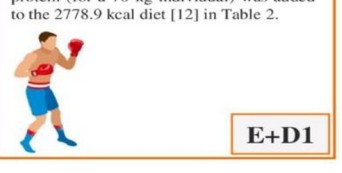

**E+D1**

Calorie restriction was performed by regarding an average of 500 kcal and reducing the daily energy requirement of each participant by 500 kcal. Therefore the diet was assigned according to ~ 2250 kcal (2778,9-500 = 2278 kcal) [14] in Table 2.

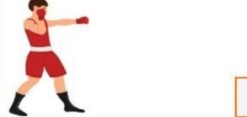

**E+D2**

**No Training & No Food-Restricted**

**Training & No Food-Restricted**

**Training & Food-Restricted**

**Fig 1. An overview of diet and exercise for boxers demonstrates.**

(Angel brand). Height measurements were taken with a sliding caliper (Holtain brand) with the boxers standing in an upright position and the scale was adjusted so that the sliding caliper touched over their heads and the length was recorded with an accuracy of ±1mm. The BMI levels of the boxers were calculated by the BMI formula, body mass (kg) / height $m^2$ (m) [12]. Before the biceps and femur bicondylar circumference measurements, the boxers wore the most appropriate clothing for the measurement. Biceps measurements were recorded via a non-elastic 7mm wide tape measure with an accuracy of ±0.1cm. During the biceps measurements, the midpoint of the distance between the upper point of the acromion on the shoulder and the elbow was marked while the boxers were standing with their forearms bent at 90 degrees. The biceps circumferences were measured from the marked point after the boxers spread their arms to the side [13]. For femoral bicondylar measurement, the athlete is seated in a chair with the knee angle at 90° flexion and the width between the medial and lateral epicondyles of the femur is calculated to an accuracy of 0.1 cm [14]. Body fat measurements and regional muscle analysis were performed using a bioimpedance device (Type-BC-418-MAIII, Tanita Body Composition Analyzer; Tanita, Tokyo, Japan) to determine left and right arm fat percentages and right and left leg fat percentages. Moreover, the amounts of muscle in the arms and legs were measured in kg.

### Measurement of isokinetic strength method

Boxers external muscle strength and internal muscle strength based on quadriceps and hamstrings were evaluated with Biodex System 4 Pro (Biodex, U.S.A) isokinetic system. Before the isokinetic strength test, boxers were asked to perform rhythmic and submaximal exercises such as jumping rope, running in place, bicycle ergometer for 10–15 minutes for warm-up. Boxers who stated that they were ready for the tests were adjusted for body positioning, joint alignment, joint rotation axis and localizations for maximum joint movement according to the user manual of the Biodex System 4 Pro. Boxers were told to perform the first test with the sturdy limb or the dominant limb to get used to the movement pattern. With these exercises, the boxers were allowed to get used to the device by using their submaximal and maximal

forces at each test speed. Accordingly, the boxers first performed 3 submaximal repetitions with knee extensors and flexors and then performed 3 maximal repetitions without a break and the mean values of peak torque (PT), work and strength were recorded. The same repetition and exercise sequence was also performed on the shoulder internal and external rotators [3].

## Statistical analysis

The data obtained were analyzed in SPSS 26.0 (IBM, Armonk, NY, USA) package software. Additionally, GraphPad Prism 8 was used to prepare the figures. Kolmogorov Smirnov Test were performed to determine the normality of the data. As a result of the analysis, the distribution was normal. According to this result, the Paired-Samples t-test was used for intra-group pre- and post-test comparisons and the One-Way ANOVA test was used for post-test comparisons between groups. The effect sizes were calculated and classified to determine the magnitude of changes among the experimental conditions as proposed by 'Cohen's *d*. An effect size classified as 0.2 was deemed small, 0.5 as medium, and 0.8 as large [15]. In addition, the effect sizes of the participants' ANOVA test results were determined according to the partial eta square ($\eta^2$) result. Accordingly, small ($\eta^2 > 0.01$), medium ($\eta^2 > 0.06$) and large ($\eta^2 > 0.14$) effect sizes [15]. The level of significance was regarded as $p < 0.05$ in all the tests.

## Results

According to Table 3, boxers in group C were 19–24 years old, in group E+D1 were 19–24 years old, and in group E+D2 were 19–22 years old. Boxers in group C had a body mass of 69.9±1.7, in group E+D1 70.5±1.8, and in group E+D2 70.2±1.6. In general, the boxers were 165–185 cm height and BMI values were 23.1–23.4. The participants practiced boxing actively for 5–9 years.

According to Fig 2A–2C, when the changes in body mass, BMI level and body fat percentage variables of the boxers were compared within groups, no significant difference was observed in C and E+D1 (p>0.05) while in the E+D2 group, a significant difference was observed in body mass (t = 1.675, Cohen's *d* = 0.61), BMI (t = 3.030, Cohen's *d* = 1.13) and body fat percentage (t = 4.953, Cohen's *d* = 1.02). In post-test comparisons among the groups, E+D2 showed notable reductions compared to the C and E+D2 group in body mass change (F = 1.078, $\eta^2 = 0.07$), BMI level (F = 1.605, $\eta^2 = 0.19$), and body fat percentage (F = 2.285, $\eta^2 = 0.1$).

According to Fig 3A and 3B, there was no significant difference in the left and right arm, and Fig 2C and 2D in the left and right leg circumference measurements of the boxers in all intra- and inter-group comparisons (p>0.05).

According to Fig 4A–4D, there was no significant difference in the left and right arm and, left and right leg muscle measurements of the boxers in the pre-post test intra-group compared (p>0.05). However, a significant difference was observed when muscle measurements from

**Table 3. Descriptive variables of the boxers.**

| Variables | C | | E+D1 | | E+D2 | |
|---|---|---|---|---|---|---|
| | Mean±Sd | Min–Max | Mean±Sd | Min–Max | Mean±Sd | Min–Max |
| Age (years) | 20.2±0.2 | 19–23 | 20.5±0.3 | 19–24 | 20.0±0.2 | 19–22 |
| Height (cm) | 175.3±1.6 | 165–185 | 174.3±1.5 | 166–184 | 176.3±1.7 | 165–183 |
| Weight (kg) | 69.9±1.7 | 68–71 | 70.5±1.8 | 68–71 | 70.2±1.6 | 68–71 |
| BMI (kg/m$^2$) | 23.3±0.47 | 23.1–23.4 | 23.2±0.39 | 23.1–23.4 | 23.3±0.53 | 23.1–23.4 |
| Sports period (years) | 7.4±1.2 | 5–9 | 7.3±1.3 | 5–8 | 7.4±1.2 | 5–9 |

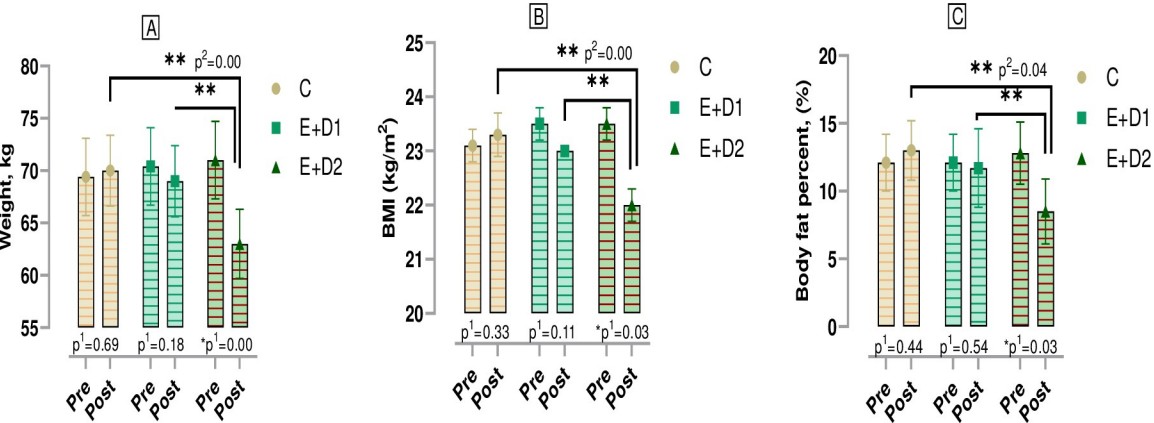

**Fig 2. Physiological parameters related to body mass, BMI and body fat percentage in boxers.** $p^1$; within group, $p^2$; posttest comparison between groups, *;$p < 0.05$. C; control, E+D1; exercise+diet1, E+D2; exercise+diet2, BMI; body mass index.

the arms (left arm; t = 5.790, Cohen's $d$ = 0.78, right arm; t = 8.209, Cohen's $d$ = 0.74) and legs (left leg; t = 3.982, Cohen's $d$ = 0.92, right leg; t = 4.133, Cohen's $d$ = 0.88) pre-post tests of boxers in the E+D2 group were compared. According to Fig 3C, when the post-test of left leg muscle measurements were compared between the groups, a significant difference was found with E+D2 compared to C and E+D1 ($p < 0.05$, F = 6.011, $\eta^2$ = 0.30). According to Fig 3C, there was a similar difference in right leg muscle measurements in the post-test comparisons between the groups. Therefore, compared to the other two groups, boxers in the E+D2 group had lower right leg muscle ratios ($p < 0.05$, F = 6.474, $\eta^2$ = 0.32).

According to Fig 5A and 5B, when the pre-post test strength measurements of the knee extensors and flexors of the boxers were compared inter-group, both quadriceps submaximal (F = 4.163, $\eta^2$ = 0.13) and quadriceps maximal (F = 5.014, $\eta^2$ = 0.16) strength output decreased

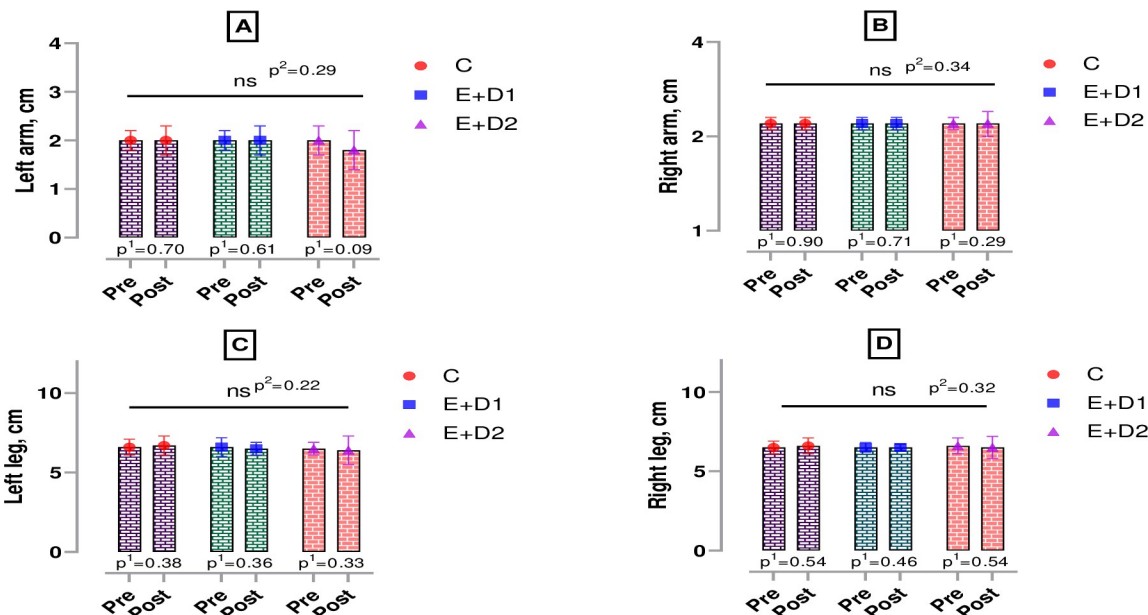

**Fig 3. Measurements of biceps and femur bicondylar somatotype circumference in boxers.** $p^1$; within group, $p^2$; posttest comparison between groups, ns; non-significant, C; control, E+D1; exercise+diet1, E+D2; exercise+diet2.

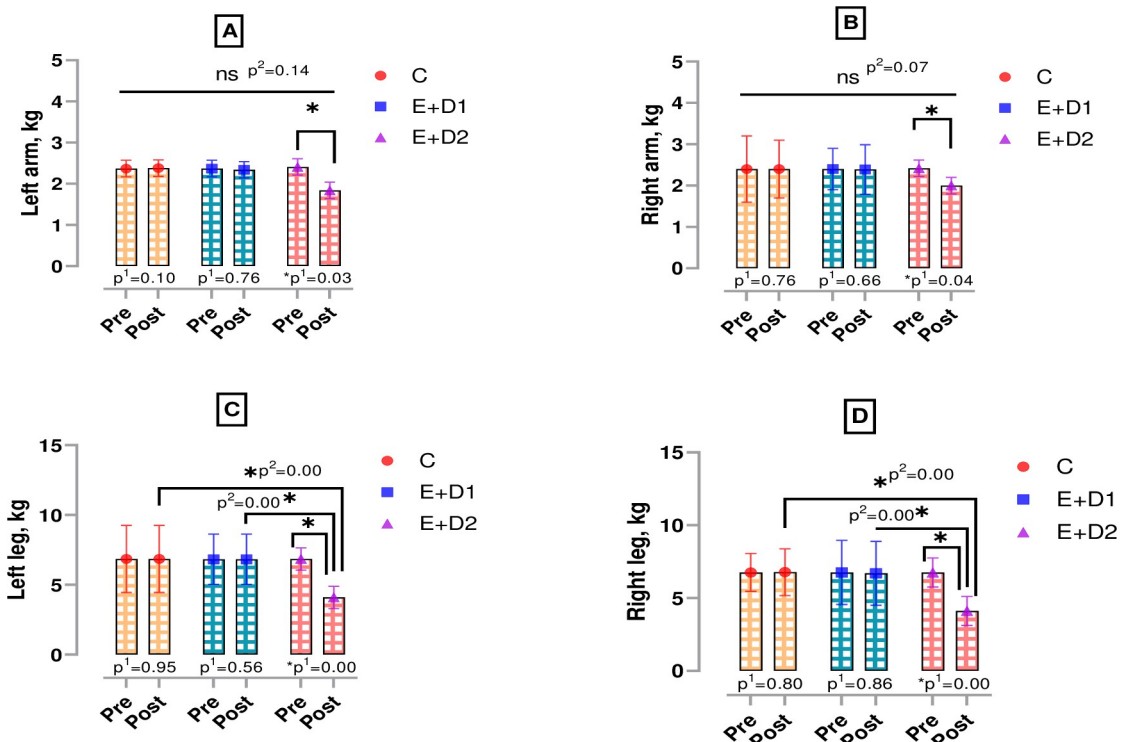

**Fig 4. Muscle measurements from the arms and legs of boxers.** $p^1$; within group, $p^2$; posttest comparison between groups, ns; non-significant, *;p<0.05, C; control, E+D1; exercise+diet1, E+D2; exercise+diet2.

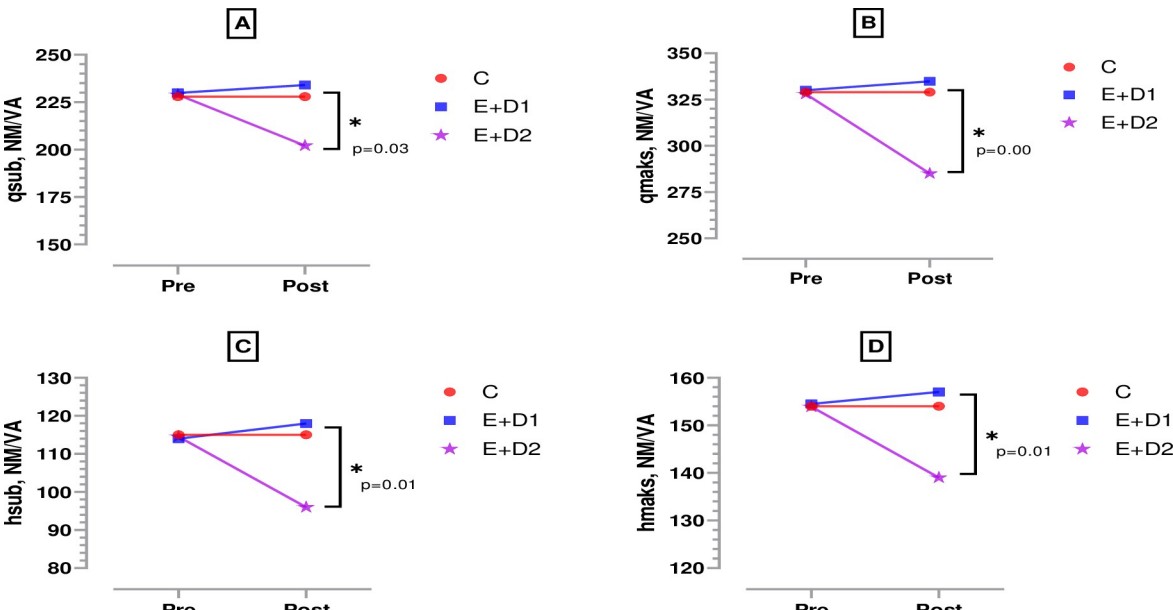

**Fig 5. Strength measurements in the knee extensors and flexors of boxers.** p; posttest measurements between groups, *;p<0.05, C; control, E+D1; exercise+diet1, E+D2; exercise+diet2, qsub; quadriceps submaximal, qmax; quadriceps maximal, hsub; hamstring submaximal, hmax; hamstring maximal.

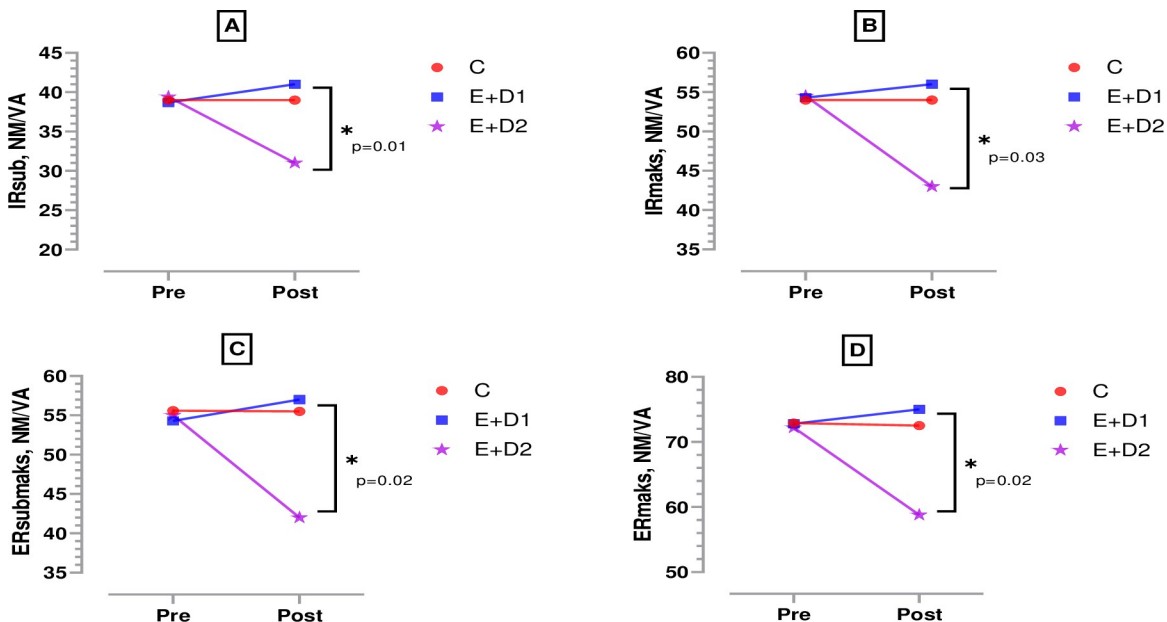

**Fig 6. Strength measurements of the shoulder internal and external rotators of boxers.** p; posttest measurements between groups, \*; p<0.05, C; control, E+D1; exercise+diet1, E+D2; exercise+diet2, IRsub; internal rotation submaximal, IRmax; internal rotation maximal, ERsub; external rotation submaximal, ERmax; external rotation maximal.

significantly in the E+D2 group. Fig 4C and 4D, both hamstring submaximal (F = 3.031, $\eta^2$ = 0.13) and hamstring maximal (F = 4.364, $\eta^2$ = 0.14) strength output decreased significantly in the E+D2 group (p<0.05). According to Fig 6A and 6B, when the pre-post test strength measurements of the shoulder internal and external rotators of the boxers were compared intergroup, both internal rotation submaximal (F = 7.631, $\eta^2$ = 0.09) and internal rotation maximal (F = 4.115, $\eta^2$ = 0.26) strength output decreased significantly in the E+D2 group. Fig 5C and 5D, both external rotation submaximal (F = 3.113, $\eta^2$ = 0.30) and external rotation maximal (F = 4.463, $\eta^2$ = 0.18) strength output decreased significantly in the E+D2 group (p<0.05).

## Discussion

At the end of the study, there was no significant difference in the C and E+D1 groups in the intra-group comparison of the body mass, BMI level and body fat percentage changes of the boxers. However, a significant difference was found in E+D2 when the group was compared via pre-tests and post-tests. In the post-test results between the groups, E+D2's body mass change, BMI level and body fat percentage decreased significantly compared to the other two groups and E+D2, who were restricted by 500 kcal daily, reached an average body mass of 63 kg. We think that the body mass loss of E+D2 in the 30 days period was due to the calorie deficiency caused by the 500 kcal daily energy restriction, low carbohydrate consumption, aerobic metabolic activity during the 90 min training period, depletion of phosphate stores and the effort to replenish these stores, increase in BMI and PA intensity. This inference can also be valid for the decrease in BMI and body fat percentage. The fact that no significant change occurred in body mass, BMI and body fat percentage of E+D1 despite being exposed to a similar training period can be explained by the protein they consumed, 1.5 g per kg per day, in addition to their adequate calorie intake.

Hirono et. al. (2022) reported in a case study with an elite boxer that body mass loss decreased body mass and muscle mass but changed the boxer's strength very slightly [16].

Zubac et al. (2018) stated that at the end of their study with 83 elite boxers who aimed to lose body mass in a short time, they had significant body mass loss [17]. However, the same study also reported that 33% of the body mass loss was muscle loss. In addition, 45% of the boxers in the study stated that they continued to do so despite knowing the negative consequences of body mass loss. It has been stated that wrestlers experience body mass loss, especially before the competition, which may have negative consequences [7, 18]. Another study that tracked body mass loss in boxers found decreased performance, decreased agility, and that boxers were more likely to be stressed and aggressive [19]. Rejic et al. (2013) reported that elite boxers undergo significant hypohydration with 5% body mass loss before competition [20]. Marugappan et al., (2022) on the other hand, stated that the athletes who lose body mass rapidly before the competition will quickly regain their former body mass after the end of the competition period [21].

The findings of this study suggest that the amount of nutrients in the diet triggers muscle catabolism and accordingly modelled body mass as well as strength, which is a basic motoric ability in the upper and lower extremities of boxers. Because muscle strength is an important physical fitness component for successful performance in sports [22, 23]. In boxing, the quality of muscle strength in both upper and lower extremities can affect the success of the athlete [24]. In particular, body mass loss tendencies added to intensified pre-competition training periods may cause injuries and significant performance decreases [25, 26]. In this context, in the examination of the circumference measurements, no significant change was observed in right and left biceps or left and right femoral bicondylar somatotype measurements in all groups. The reasons for these changes observed between the groups may be muscle catabolism due to inadequate protein intake, insufficient daily energy consumption and weight loss in a short time. Literature information on the subject supports our study. In their study on jockeys, Wilson et al (2013) found that the group with rapid weight loss due to malnutrition had a decrease in leg muscle strength compared to the control group [27]. In another study examining athletes interested in mixed martial arts, it was reported that the athlete whose nutrition was regulated showed more positive results in the average strength parameter compared to the other [28]. In addition, in a meta-analysis study conducted in 2017, 1863 participants were examined and it was proven that protein intake in the required amount for the body increased muscle mass and muscle strength parameters [29]. Recently, Karatoprak et al. (2022) evaluated the effects of high (2.5–2.7 g/kg) and recommended (1.8–2.0 g/kg) amounts of protein consumption on body composition and performance output in young male individuals performing strength training for bodybuilding, and found that muscle mass increased in both groups [30]. This increase also improved the participants' performances in the squat, sit-up, push-up and pull-up tests compared to the pre-test. In the post-test results between the groups, the performance of the participants in the high-protein group improved more than the participants who received the recommended level of protein.

When we compared the strength measurements of submaximal and maximal knee extensors and flexors, quadricep and hamstring strength production of E+D2 was significantly decreased compared to the other two groups. In the comparison of the shoulder internal rotators (subscapular, pectoralis major, latissimus dorsi, teres major and anterior deltoid) and shoulder external rotators (infraspinatus, teres minor and posterior deltoid), the force production of E+D2 was significantly reduced compared to the other two groups. Although this was not significant in E+D1 in both cases, it was in the form of a small increase. In K, it was discovered to remain constant. Aydos (1996) investigated the effects of body mass loss of at least 5% per individual within 48 hours on some physiological performance change processes with 17 volunteer wrestling athletes and reported that body mass loss negatively affected general endurance, basic strength, quick strength and aerobic capacity at different levels [31]. Likewise,

in a study in which calorie restriction was applied to wrestlers to reduce their body mass, it was stated that body mass loss led to power losses in wrestlers before the competition and also caused injuries [32]. It was found that muscle anabolism was negatively affected in boxers who performed resistance training with nutrient restriction and this decreased the athletic performance of boxers [33]. In a study investigating the effects of pre-competition body mass loss on body wight, fat-free mass, sum of skinfolds and body fat in elite athletes, no significant difference was found at baseline. However, it was observed that body wight, fat-free mass, sum of skinfolds and body fat decreased significantly in the group that restricted food two days before the competition, and increased to baseline levels one week after the competition [34].

## Conclusion

In conclusion, body mass loss attempts to gain a body mass advantage before the competition period may seem to yield positive results in terms of BMI, body mass and fat percentage but may negatively affect power output and thus, performance quality. Within this context, athletes should maintain their competitive body mass throughout the season while standing away from body mass advantage expectations and should avoid nutritional restrictions that may affect their strength performance. Therefore, athletes should be provided with adequate calorie and nutrient intake in the pre-competition period.

## Strengths, limitations and suggestions for future research

The limitations of this research are that it was conducted only with local and male boxers. There were no measurement records on total water loss and the food consumption of boxers during leisure time was not followed. Moreover, there were no possibilities to follow the boxers' achievements in the championships. We did not have authorisation for this. Therefore we avoided the statement that it definitely increased the athletes' competition performance. Finally, it would be valuable for future research to follow the performance or success of athletes during competition.

## Supporting information

**S1 Data.**
(XLSX)

## Author Contributions

**Conceptualization:** Yavuz Yasul.

**Data curation:** Yavuz Yasul, Faruk Akçınar.

**Formal analysis:** Yavuz Yasul, Faruk Akçınar.

**Methodology:** Faruk Akçınar, Muhammet Enes Yasul.

**Resources:** Luca Paolo Ardigò.

**Software:** Muhammet Enes Yasul.

**Visualization:** Ahmet Kurtoğlu, Georgian Badicu.

**Writing – original draft:** Ahmet Kurtoğlu, Özgür Eken, Georgian Badicu.

**Writing – review & editing:** Özgür Eken, Georgian Badicu, Luca Paolo Ardigò.

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
