## [Decision Letter · Decision Letter 0]

23 Jan 2024

PONE-D-23-36041Effects of Short-Term Pre-Competition Weight Loss on Certain Physiological Parameters and Strength Change in Elite BoxersPLOS ONE

Dear Dr. Badicu,

Thank you for submitting your manuscript to PLOS ONE. After careful consideration, we feel that it has merit but does not fully meet PLOS ONE’s publication criteria as it currently stands. Therefore, we invite you to submit a revised version of the manuscript that addresses the points raised during the review process. Please submit your revised manuscript by Mar 08 2024 11:59PM. If you will need more time than this to complete your revisions, please reply to this message or contact the journal office at plosone@plos.org. Please include the following items when submitting your revised manuscript:A rebuttal letter that responds to each point raised by the academic editor and reviewer(s). You should upload this letter as a separate file labeled 'Response to Reviewers'.A marked-up copy of your manuscript that highlights changes made to the original version. You should upload this as a separate file labeled 'Revised Manuscript with Track Changes'.An unmarked version of your revised paper without tracked changes. You should upload this as a separate file labeled 'Manuscript'.If applicable, we recommend that you deposit your laboratory protocols in protocols.io to enhance the reproducibility of your results. Protocols.io assigns your protocol its own identifier (DOI) so that it can be cited independently in the future. For instructions see: https://journals.plos.org/plosone/s/submission-guidelines#loc-laboratory-protocols. Additionally, PLOS ONE offers an option for publishing peer-reviewed Lab Protocol articles, which describe protocols hosted on protocols.io. Read more information on sharing protocols at https://plos.org/protocols?utm_medium=editorial-email&utm_source=authorletters&utm_campaign=protocols.

We look forward to receiving your revised manuscript.

Kind regards,

Jovan Gardasevic

Academic Editor

PLOS ONE

Journal Requirements:

5. We notice that your supplementary figures are uploaded with the file type 'Other'. Please amend the file type to 'Supporting Information'. Please ensure that each Supporting Information file has a legend listed in the manuscript after the references list.

Additional Editor Comments:

I have read it carefully, I feel that certain corrections should be made to make it ready for publication. Focus on the reviewers' objections and try to correct their comments. 

Reviewers' comments:

Reviewer's Responses to Questions

**Comments to the Author**

1. Is the manuscript technically sound, and do the data support the conclusions?

Reviewer #1: Yes

Reviewer #2: Yes

2. Has the statistical analysis been performed appropriately and rigorously? 

Reviewer #1: Yes

Reviewer #2: I Don't Know

3. Have the authors made all data underlying the findings in their manuscript fully available?

Reviewer #1: Yes

Reviewer #2: Yes

4. Is the manuscript presented in an intelligible fashion and written in standard English?

Reviewer #1: Yes

Reviewer #2: Yes

5. Review Comments to the Author

Reviewer #1: Boxers lose weight and lose strength with a restrictive diet.

This study did not convince me of its justification.

The authors did what is already known.

It would be useful for practice if the authors gave us an answer to the following: Is the strength they possess after the experimental program still higher than those who do not remove weight?

However, we do not have that answer.

The technical manuscript is extremely well prepared. I have no complaints from that side.

I read a lot of interesting information and learned something while doing this review. However, I remain of the opinion that this manuscript did not reach the quality for the Plos One rating journal.

Reviewer #2: Thank you for the opportunity to review the manuscript titled “Effects of Short-Term Pre-Competition Weight Loss on Certain Physiological Parameters and Strength Change in Elite Boxers” (PONE-D-23-36041). The authors aim to evaluate the change in body composition of boxers whose pre-competition diets were under control and how this change affects both knee extensors and flexors and shoulder internal and external rotators muscle strength. In the study, it was assumed that E+D2 would have an optimization in body composition and a reduced power output, whereas in E+D1, the two parameters would be fixed or in an increasing tendency.

For transparency, I have limited understanding and little experience with boxing and practical activities applied by the authors of this study; however, the application of the theoretical knowledge into the practice and comments related to these issues, as well as overall review, such as part of results and discussion should be considered.

The results are written clearly and supported by the appropriate tables and figures in text. The authors also provide a great deal of analysis and data in appendices to the manuscript which is appreciated. The results are appropriately discussed, with the authors drawing on relevant literature and are careful with their interpretations given the quality of the evidence used in the review. The authors appropriately acknowledge the limitations of this work. The authors are to be commended on an interesting manuscript, and a valuable piece of work that treats separate categories of the boxers and assessed the association between different approaches of diet and training interventions.

I would recommend to the authors the followings:

[1] Exclude abbreviation in abstract that are not well-known e.g. describe the groups of boxers clearly etc;

[2] Do not put unnecessary information in abstract as the abstract has to contain just the most important information that will persuade potential readers to read the whole manuscript e.g. “The data obtained were analyzed in SPSS 22.0 package software” – this sentence is absolutely unnecessary etc.;

[3] The method section is too long and it is hard to read the manuscript, so I recommend to use some protocol that is already published and just cite it with some highlighted modifications; This can be also used from PlosOne protocol section or protocols.io;

[4] From the reason the limitations are mentioned but did not explained briefly, I do believe it is necessary to add recommendations for the further studies and share some experiences from this study to help researchers who will replicate this study in setting up the study designed and avoid some eventual methodological issues;

[5] the reference list should be revised and updated by full meta data.

My comments to the authors are all relatively minor and are primarily there to perhaps improve the readability of the manuscript.

6. PLOS authors have the option to publish the peer review history of their article (what does this mean?). If published, this will include your full peer review and any attached files.

Reviewer #1: No

Reviewer #2: **Yes: **Stevo Popovic

---

## [Author Response · Author response to Decision Letter 0]

14 Mar 2024

Academic Editor

About Journal Requirements:

Answer: updated, thank you!

Please provide additional details regarding participant consent. In the ethics statement in the Methods and online submission information, please ensure that you have specified (1) whether consent was informed and (2) what type you obtained (for instance, written or verbal, and if verbal, how it was documented and witnessed). If your study included minors, state whether you obtained consent from parents or guardians. If the need for consent was waived by the ethics committee, please include this information. If you are reporting a retrospective study of medical records or archived samples, please ensure that you have discussed whether all data were fully anonymized before you accessed them and/or whether the IRB or ethics committee waived the requirement for informed consent. If patients provided informed written consent to have data from their medical records used in research, please include this information.

Answer: Dear Editor, Our research did not involve under-aged children, so we did not need to obtain parental consent. The process was organised with a consent form in which all data will be completely anonymised and the ethics committee was informed before accessing the data. Informed written consent was obtained from the participants for the use of the data in their records in the research. Also separate written consent form was obtained for each group and the method section (study design) was presented.

Answer: Dear Editor, we have presented the ethical statement of the research in the method section. As you have stated, we have not re-presented the information about our ethical statement in any other section.

Answer: updated, thank you!

5. We notice that your supplementary figures are uploaded with the file type 'Other'. Please amend the file type to 'Supporting Information'. Please ensure that each Supporting Information file has a legend listed in the manuscript after the references list.

Answer: updated, thank you!

Answer: Dear Editor, we have reviewed our reference list according to your journal rules and tried to update the references that were found to be inappropriate and incorrect, and tried to show the changes made with the red font "Revised Manuscript with Track Changes’’. In addition, the references have not been replaced and no references have been removed from the main text.

Reviewer #1:

1. It would be useful for practice if the authors gave us an answer to the following: Is the strength they possess after the experimental program still higher than those who do not remove weight?

Answer: Thanks for your professional review work on our article. We agree with your point of view but there were no possibilities to follow the boxers' achievements in the championships. We did not have authorisation for this. Therefore we avoided the statement that it definitely increased the athletes' competition performance. In addition, one of the important limitations of the research included a follow-up and evaluation before the championships. However, we were able to report some of our observations about the change in performance. Although there was no statistically significant difference between the C group and the E+D1 group, quadriceps submaximal and maximal strength development and hamstring submaximal and maximal strength development of the E+D1 group were higher than the C group. Similar situation was observed in shoulder internal and external rotators as well as arms and legs measurements. This situation suggests that especially for boxers, rapid weight loss may be a disadvantage for them, on the contrary, they may prevent loss of strength in a planned program with adequate caloric intake and protein support instead of short-term dietary restrictions while dropping weight. Therefore, we consider this research to be important in terms of demonstrating the value of controlling the training and nutrition periods of boxers with adequate calorie consumption and protein intake.

Reviewer #2: 

1. Exclude abbreviation in abstract that are not well-known e.g. describe the groups of boxers clearly etc;

Answer: Thanks for your professional review work on our article. We agree with your point of view, we've describe the groups of boxers and abstract section accordingly, corresponding changes have been highlighted in red in the revised version.

2. Do not put unnecessary information in abstract as the abstract has to contain just the most important information that will persuade potential readers to read the whole manuscript e.g. “The data obtained were analyzed in SPSS 22.0 package software” – this sentence is absolutely unnecessary etc.;

Answer: Thank you for your generous comments. We very much valuable your view that "The summary should contain only the most important information, do not put unnecessary information in the summary". We have made the appropriate changes and they are highlighted in red in the revised text.

3. The method section is too long and it is hard to read the manuscript, so I recommend to use some protocol that is already published and just cite it with some highlighted modifications

Answer: Thank you very much for your professional comments about the method. You are correct that it is long and difficult to read. Considering your valuable guidance, we have tried to present the diet and exercise part with a summarising figure and presented the protocols that we have taken as reference. We have tried to present the diet content and exercise method as in the research, since we have not come across such an approach and research on boxers before, and also in order to follow the effects of the boxers' pre-competition nutrition and exercise intensity.

4. From the reason the limitations are mentioned but did not explained briefly, I do believe it is necessary to add recommendations for the further studies and share some experiences from this study to help researchers who will replicate this study in setting up the study designed and avoid some eventual methodological issues;

Answer: Thanks so much for your useful comments. We apologize for the confusion caused by our unclear expression. Thanks to you, we have recognised our shortcoming regarding limitations and have done our best to overcome it and make suggestions for future research.

5. The reference list should be revised and updated by full meta data.

Answer: We are thankful that we have updated the deficiencies in the reference list, which we have overlooked and which you are absolutely right, and presented them again for your evaluation.

Thank you!

---

## [Decision Letter · Decision Letter 1]

6 May 2024

PONE-D-23-36041R1Effects of Short-Term Pre-Competition Weight Loss on Certain Physiological Parameters and Strength Change in Elite BoxersPLOS ONE

Dear Dr. Badicu,

Thank you for submitting your manuscript to PLOS ONE. After careful consideration, we feel that it has merit but does not fully meet PLOS ONE’s publication criteria as it currently stands. Therefore, we invite you to submit a revised version of the manuscript that addresses the points raised during the review process.

**I agree with the reviewer's decision on minor revision of the manuscript, please submit the manuscript again for review after that correction.**

We look forward to receiving your revised manuscript.

Kind regards,

Jovan Gardasevic

Academic Editor

PLOS ONE

Journal Requirements:

Reviewers' comments:

Reviewer's Responses to Questions

**Comments to the Author**

1. If the authors have adequately addressed your comments raised in a previous round of review and you feel that this manuscript is now acceptable for publication, you may indicate that here to bypass the “Comments to the Author” section, enter your conflict of interest statement in the “Confidential to Editor” section, and submit your "Accept" recommendation.

Reviewer #2: All comments have been addressed

Reviewer #3: (No Response)

2. Is the manuscript technically sound, and do the data support the conclusions?

Reviewer #2: Yes

Reviewer #3: Yes

3. Has the statistical analysis been performed appropriately and rigorously? 

Reviewer #2: Yes

Reviewer #3: Yes

4. Have the authors made all data underlying the findings in their manuscript fully available?

Reviewer #2: Yes

Reviewer #3: Yes

5. Is the manuscript presented in an intelligible fashion and written in standard English?

Reviewer #2: Yes

Reviewer #3: Yes

6. Review Comments to the Author

**Reviewer #2:** Thank you very much for accepting all my recommendations. I have no further requests and I recommend that this manuscript can be accepted.

**Reviewer #3:** very good study. the authors have decided on a topic that is very interesting and current. I have to praise the methodological setting of the work, because the method of research is presented very clearly and in detail.

my only objection is of a technical nature:

- when you show figure 1, you put its title above the graph, and for the others (figure 2, figure 3...) you put the title below the graph. It must be identical, as prescribed by the standard.

7. PLOS authors have the option to publish the peer review history of their article (what does this mean?). If published, this will include your full peer review and any attached files.

Reviewer #2: **Yes: **Stevo Popovic

Reviewer #3: No

---

## [Author Response · Author response to Decision Letter 1]

7 May 2024

Reviewer #3: very good study. the authors have decided on a topic that is very interesting and current. I have to praise the methodological setting of the work, because the method of research is presented very clearly and in detail.

my only objection is of a technical nature:

- when you show figure 1, you put its title above the graph, and for the others (figure 2, figure 3...) you put the title below the graph. It must be identical, as prescribed by the standard.

Response: Adjusted, thank you very much. 

Best regards!

---

## [Editor Report · Decision Letter 2]

9 May 2024

Effects of Short-Term Pre-Competition Weight Loss on Certain Physiological Parameters and Strength Change in Elite Boxers

PONE-D-23-36041R2

Dear Mr. Georgian Badicu,

We’re pleased to inform you that your manuscript has been judged scientifically suitable for publication and will be formally accepted for publication once it meets all outstanding technical requirements.

Kind regards,

Jovan Gardasevic

Academic Editor

PLOS ONE
---

## [Editor Report · Acceptance letter]

13 May 2024

PONE-D-23-36041R2 

PLOS ONE

Dear Dr. Badicu, 

I'm pleased to inform you that your manuscript has been deemed suitable for publication in PLOS ONE. Congratulations! Your manuscript is now being handed over to our production team.

Kind regards, 

on behalf of

Dr. Jovan Gardasevic 

Academic Editor

PLOS ONE